# Elephant Black Garlic’s Beneficial Properties for Hippocampal Neuronal Network, Chemical Characterization and Biological Evaluation

**DOI:** 10.3390/foods12213968

**Published:** 2023-10-30

**Authors:** Javiera Gavilán, Claudia Mardones, Gabriela Oyarce, Sergio Triviño, Nicole Espinoza-Rubilar, Oscar Ramírez-Molina, Claudia Pérez, José Becerra, Patricio Varas, Robinson Duran-Arcos, Carola Muñoz-Montesino, Gustavo Moraga-Cid, Gonzalo E. Yévenes, Jorge Fuentealba

**Affiliations:** 1Departmento de Fisiologia, Facultad de Ciencias Biologicas, Universidad de Concepcion, Concepción P.O. Box 160-C, Chile; javieragavilan@udec.cl (J.G.); nicespinoza@udec.cl (N.E.-R.); oramirezm@udec.cl (O.R.-M.); rduran2016@udec.cl (R.D.-A.); carmunoz@udec.cl (C.M.-M.); gumoraga@udec.cl (G.M.-C.); 2Facultad de Farmacia, Universidad de Concepcion, Concepción P.O. Box 160-C, Chile; cmardone@udec.cl; 3Laboratorio de Quimica de Productos Naturales, Facultas de Ciencias Naturales y Oceanograficas, Universidad de Concepcion, Concepción P.O. Box 160-C, Chile; gabrielaoyarce@udec.cl (G.O.); strivino1984@gmail.com (S.T.); claudiaperez@udec.cl (C.P.); jbecerra@udec.cl (J.B.); 4Melimei Agricultural Company, Ancud PC 5710325, Chile; pvaras@melimei.cl; 5MinusPain, Facultad de Ciencias Biológicas, Universidad de Concepcion, Concepción P.O. Box 160-C, Chile

**Keywords:** *A. ampeloprasum*, fresh garlic, aged garlic, sulphurated compounds, antioxidant activity, synaptic function

## Abstract

Garlic has been used for decades as an important food and additionally for its beneficial properties in terms of nutrition and ancestral therapeutics. In this work, we compare the properties of fresh (WG) and aged (BG) extract obtained from elephant garlic, harvested on Chiloe Island, Chile. BG was prepared from WG with a 20-day aging process under controlled temperature and humidity conditions. We observed that in BG, compounds such as diallyl disulfide decrease, and compounds of interest such as 5-hydroxymethylfurfural (69%), diallyl sulfide (17%), 3H-1,2-Dithiole (22%) and 4-Methyl-1,2,3-trithiolane (16%) were shown to be increased. Using 2,2-diphenyl-1-picrylhydrazyl (DPPH, BG: 51 ± 5.7%, WG: 12 ± 2.6%) and 2,20-azino-bis-(3-ethylbenzothiazoline-6 sulfonate) diammonium salt (ABTS, BG: 69.4 ± 2.3%, WG: 21 ± 3.9%) assays, we observed that BG possesses significantly higher antioxidant activity than WG and increased cell viability in hippocampal slices (41 ± 9%). The effects of WG and BG were shown to improve the neuronal function through an increased in intracellular calcium transients (189 ± 4%). In parallel, BG induced an increase in synaptic vesicle protein 2 (SV-2; 75 ± 12%) and brain-derived neurotrophic factor (BDNF; 32 ± 12%) levels. Thus, our study provides the initial scientific bases to foster the use of BG from Chiloe Island as a functional food containing a mixture of bioactive compounds that may contribute to brain health and well-being.

## 1. Introduction

Common garlic (*Allium sativum*) has been employed over many years for culinary and medicinal purposes [1]. The properties of garlic have been studied in more than 3000 publications, which have reported a large variety of potential beneficial effects for health from diverse garlic preparations [2]. The most studied cellular effects have been related to the intracellular calcium modulation [3], while the beneficial cardiovascular effects have been the most frequently reported [4]. The wide number of garlic beneficial effects have been associated with compounds of the garlic bulb. A main focus of attention has been placed on the allyl-cysteine derivates [5], which are among the main components of common garlic [6].

More recent reports have highlighted the beneficial properties of aged garlic (AGE, also known as black garlic), which is a preparation obtained through a sequential oxidative process. These steps induce changes in the organoleptic properties and chemical composition of the garlic clove, which have fostered the use of AGE in modern culinary preparations [7]. The properties of AGE preparations have been evaluated in biochemical, cellular and behavioral models of pathological conditions [8]. For example, a clinical study in patients with metabolic syndrome (MetS) revealed that oral AGE treatments (2400 mg/day, for 52 weeks) reduced coronary plaque volume and adverse cardiovascular events [9]. These investigations have promoted the consumption of AGE and the commercialization of AGE-based supplements, such as AGE powder pills, oil capsules and extracts.

AGE is prepared from fresh garlic cloves and does not involve the addition of synthetic compounds. The process typically includes a combination of controlled temperature and humidity for a variable period of time [10]. The manufacturing time depends on the humidity and temperature conditions. The most usual temperature range used is 40–90 °C, while the relative humidity employed lies between 80% and 95%. AGE is ready for consumption after a period between 10 and 40 days [11]. AGE preparation generates significant changes in the profile of secondary metabolites. Interestingly, the concentration of bioactive compounds, such as phenols, flavonoids, pyruvate, furans, thiosulfate and organosulfur compounds (mainly S-allylcysteine (SAC) and S-allylmercaptocysteine (SAMC)), is increased [12]. Additional molecules similar to SAC, such as diallyl sulfide (DAS), dimethyl trisulfide (DMTS) and allyl-methyl- trisulfide (AMTS), can be also produced as part of the aging process [6].

The beneficial effects of AGE have been related to antioxidant properties. SAC has been postulated as a key molecule to explain the beneficial effects of AGE. For instance, biochemical assays have shown the activity of SAC as an effective scavenger of reactive oxygen species, including superoxide anion, hydroxyl radical, peroxynitrite anion and hydrogen peroxide [13]. Additional studies have shown that SAC also modulates the expression and the activity of several enzymes involved in the cell physiology, such as NADPH-quinone oxidoreductase, nitric oxide synthase, cyclooxygenase-2 and xanthine oxidase [14]. Because oxidative stress has been extensively linked to neurodegenerative disorders, other lines of research have investigated whether AGE and SAC can exert neuroprotective and neurotrophic effects [5,15]. For instance, cellular studies showed that AGE (1 mg/mL) enhanced the survival of cultured hippocampal neurons [16]. Similar experiments determined that ten different organosulfur compounds derived from AGE, including SAC and SMAC, were able to exert positive effects on neuronal survival [5]. In addition, these authors showed that AGE or SAC alone generated an increase in the number of branching points per axon in cultured hippocampal neurons [5,16]. Regarding neuroprotection, SAC was shown to partially protect against the neuronal death induced by micromolar concentrations of amyloid β-peptide (Aβ) in cultured neurons [17]. Other experiments showed that the neurotoxicity induced by Aβ together with the activation of N-methyl-D-aspartate receptor (NMDAR) was mitigated by micromolar concentrations of SAC administered to organotypic hippocampal slices [18]. Regarding the molecular and cellular mechanisms associated to the neuroprotective and neurotrophic effects, biochemical studies have proposed that AGE and/or SAC changes the neuronal gene expression [19] and improves endoplasmic reticulum stress [20]. Although these and others have offered evidence strongly pointing to both AGE and SAC exerting neuroprotective and neurotrophic effects, little is known about the effects of AGE in terms of the synaptic function. Moreover, whether these treatments modify the expression of neurotrophins has not been explored. Such research is critical to establish the potential of AGE and its chemical components as neuroactive compounds of natural origin capable of acting against neurodegenerative diseases.

We focus our research on an elephant garlic native to Chiloe Island (Patagonia, Chile) that belongs to the species *Allium ampeloprasum var. ampeloprasum*. This species has been cultivated on Chiloe Island for centuries and has not been crossed with other species of garlic. Therefore, even though there are other studies of *Allium ampeloprasum* cultivated in other countries [21,22], it is important for us to evaluate the chemical composition and potential bioactive properties of this native garlic, because growing conditions and environmental factors can cause changes in its chemical composition. In addition, here, we set out a molecular, cellular and functional characterization of the potential neuroactive properties of fresh garlic (white garlic, WG) and aged garlic (black garlic, BG) of *Allium ampeloprasum* from Chiloe Island.

## 2. Materials and Methods

### 2.1. Chemicals and Reagents

DPPH, ABTS, 2,4,6-Tris(2-pyridyl)-s-triazine (TPTZ), 6-hydroxy-2,5,7,8-tetramethyl2-carboxylic acid (Trolox), hydrochloric acid (HCl) and gallic acid were obtained from Sigma-Aldrich (St. Louis, MO, USA). Sodium carbonate (Na_2_CO_3_), methanol (MeOH), potassium dihydrogen phosphate (KH_2_PO_4_), sodium chloride (NaCl), sodium hydroxide and HCl were purchased from J. T. Baker Inc. (Phillipsburg, NJ, USA). All reagents and solvents used were of analytical or high-performance liquid chromatography (HPLC) grade.

### 2.2. Preparation of Aged Garlic Extract

The preparation of BG from fresh elephant garlic cloves was performed by the agricultural company Melimei (Manao, Chiloe Island, Chile). Briefly, fresh garlic cloves of the *Allium ampeloprasum* species were subjected to an aging process, where a temperature of 70 °C and humidity of 90% were applied, for a period of 20 days [11,23].

To prepare the garlic extracts, 50 g of chopped WG or BG cloves was subjected to maceration in MeOH at HPLC grade (≤99.9% purity) under magnetic stirring for 48 h at room temperature (ca. 19 °C). Subsequently, the extract was centrifuged at 4000× *g* for 20 min, and the supernatant was filtered through polycarbonate 0.45 μm pore size membranes (Merck Chemicals, Darmstadt, Germany). The extraction procedure was repeated three times, and supernatants were pooled together. All extractions were performed in triplicate, avoiding light exposure. The extracts were concentrated in a rota-evaporator (Heidolph WB 2001) and then lyophilized. The extracts were stored in the dark at 4 °C. To perform the biological tests, the extracts were suspended in DPBS 1× (Dulbecco’s Phosphate Buffered Saline, Gibco, Grand Island, NY, USA) to the desired concentration.

### 2.3. Acid Hydrolysis of Extracts

The white and black garlic clove crude extracts from the methanolic extraction were subjected to acid hydrolysis. The extracts were re-suspended in distilled water and a conventional defatting process with hexane was performed for the defatted samples followed by ethyl acetate (AcOEt) extraction. After filtration, AcOEt was evaporated, lyophilized and stored at −50 °C.

Briefly, the acid hydrolysis of the extract (AcOEt) was made by adding 10 mL of 5% HCl to a solution containing 2.50 mg of sample in MeOH (10 mL), and the mixture was subjected to refluxing for 8 h. The reaction mixture was neutralized with saturated Na_2_CO_3_ and extracted with AcOEt (2 × 5 mL) to give an organic fraction containing the aglycone part. The organic phase was concentrated. The organic extracts were washed with water, dried (anhydrous Na_2_SO_4_) and concentrated using a rotary evaporator. The extract was stored in the dark at 4 °C.

### 2.4. Headspace Solid-Phase Microextraction (HS-SPME)

Adsorption of sulfur compounds (volatiles) was performed in triplicate with a mass of 1.5 g of fresh *Allium ampeloprasum* plant material. The samples were placed in hermetically sealed vials. The HS-SPME technique was used for the adsorption of the compounds using a 30 µm PDMS (polydimethylsiloxane) fused silica 24 Ga SUPELCO fiber, coupled to a manually adjustable SUPELCO SPME fiber holder. The samples were stabilized and conditioned in a temperature-controlled bath (40 °C, 20 min) with constant agitation. The internal temperature of the bath was constantly monitored by means of a VWR thermocouple with double reading ±0.1 °C. The conditioning time of the samples took 10 min, and the adsorption of volatiles took 10 min; then, the compounds adsorbed on the fiber were desorbed on the liner of a gas chromatograph coupled with a mass spectrometer for 10 min. The profiles of volatile compounds were obtained using gas chromatography–mass spectrometry (GC-MS), where the applied temperature program starts at 50 °C for 5 min, then a heating ramp is started at 13 °C/min until reaching 275 °C and this temperature is maintained for 32 min.

### 2.5. Gas Chromatography–Mass Spectrometry (GC-MS)

GC-MS was used to analyze and compare the components in WG and BG clove samples. The analysis of underivatized compounds was performed on a gas chromatograph (Agilent 5975, Technologies, Santa Clara, CA, USA) fitted with an HP-5MS capillary column (30 m × 0.25 mm internal diameter, 0.25 μm film thickness), and high-purity helium (99.9%) was used as a carrier gas at a constant flow rate of 1 mL/min. The temperature program was: 5 min hold at 100 °C, 100–275 °C at 13 °C/min and 32 min hold at 275 °C. The ionization voltage was 70 eV, there was a scanning mass range *m*/*z* of 50–550 Dalton (Da) and the injector temperature was 250 °C. The samples were diluted with AcOEt, filtered and 1 μL was injected using a syringe into the injector with a split ratio of 30:1. The components were identified by matching mass spectra to records in the NIST17 (NIST/EPA/NIH MASS 2017 Spectral Library) and comparing the obtained spectra with those reported in the literature. NIST17 collects representative compounds of the scaffolds referenced. The percentage composition of the crude extract constituents was expressed as a percentage (%) by peak area. The identification structure of a compound was tentatively assigned when the overlap with the database exceeded 95%. Scaffolds of unidentified compounds were proposed according to the highest index like those listed in the database. Internal normalization was performed to compare the chromatographic profiles of white and black garlic cloves.

### 2.6. Antioxidant Activity

#### 2.6.1. 2,2-Diphenyl-1-picrylhydrazyl (DPPH)

The free radical scavenging activity of the extracts, based on the scavenging activity of the stable DPPH free radical, was determined using the method described by Brand-Williams and coworkers [24] with slight modifications. BG and WG extracts were resuspended in methanol solvent and a sample of each extract (20 µL) was added to 0.6 mM DPPH dissolved in MeOH solution (180 µL). After incubating the solution at room temperature in the dark for 30 min, the absorbance was measured at 515 nm, and the radical scavenging activity was expressed as a percent inhibition: DPPH inhibition (%) = ([AC − AS]/AC) × 100, where AC was the absorbance of the control (blank), and AS was the absorbance of the extract. Trolox (6-hydroxy-2,5,7,8-tetramethylchroman-2-carboxylic acid) was used as a standard to construct the calibration curve. Stock standard solutions were prepared in MeOH. Working solutions were prepared via dilution and, to avoid degradation, were stored at 4 °C in the dark. The calibration line was constructed with linear regression using five concentrations. The determination coefficient was 0.995 for the linear range of 0.2–3 mM. Measurements were performed in triplicate. Absorbance values were corrected for radical decay using blank solutions.

#### 2.6.2. Estimation of Trolox Equivalent Antioxidant Capacity (TEAC) Using 2,20-Azino-bis-(3-ethylbenzothiazoline-6 sulfonate) Diammonium Salts (ABTS)

ABTS was dissolved in water to a 3.5 mM concentration. ABTS radical cation (ABTS^•+^) was produced by reacting ABTS stock solution with 1.225 mM potassium persulfate (1:1) and allowing the mixture to stand in the dark at room temperature for 12–16 h before use. After storage in the dark for 16 h, the radical cation solution was further diluted in ethanol until the initial absorbance value of 0.7 ± 0.05 at 750 nm was reached. To 20 µL of each stock solution previously prepared, 180 µL of the radical solution was added and the absorbance was measured at 750 nm after 40 min in the dark. Trolox was used as a standard to construct the calibration curve. The results were thus expressed as TEAC values [25]. Measurements were performed in triplicate.

### 2.7. Cell Culture

PC-12 cells of adrenal gland pheochromocytoma (ATCC, Manassas, VA, USA) were cultured in DMEM (Dulbecco’s Modified Eagle Medium) with fetal bovine serum (FBS, 5%), horse serum (HS, 5%), penicillin (100 U/mL), streptomycin (100 μg/mL) and L-glutamine (2 mM). The cells were incubated under standard conditions (37 °C, 5% CO_2_). The cells were plated (100,000 cells/well) and used 24 h after.

Primary embryonic hippocampal cultures (E18) were seeded at 300,000 cells/mL on coverslips with poly-L-lysine (Trevigen). Cultures were maintained for 24 h in minimal essential medium (MEM; Gibco) supplemented with HS (10%), L-glutamine (2 mM) and DNAse (4 mg/mL). Then, the culture medium was replaced by another medium composed of: MEM (Gibco), HS (2%), FBS (2%) and N3 (0.5%). Composition of N3: putrescine (4 mg/mL), BSA (1 mg/mL), insulin (1.25 mg/mL), corticosterone (4 µg/mL), TH3 (2 µg/mL), progesterone (1.25 µg/mL) and sodium selenite (1 µg/mL). Experiments were performed between 10 and 11 days of in vitro incubation (DIV). Throughout this time, the cultures were maintained in a thermoregulated incubator (37 °C, 5% CO_2_). C57BL/6 mice (20 females) were anesthetized by using CO_2_ inhalation and sacrificed via cervical dislocation.

### 2.8. Preparation of Hippocampal Slices

Mice (C57BL/6 mice, females, 3–4 months old, 20–25 g) were anesthetized with isoflurane, and brains were quickly removed from the skull and deposited in a cutting solution (Cutting Solution: NaCl 120 mM, KCl 1 mM, CaCl_2_ 0.5 mM, NaHCO_3_ 26 mM, MgSO_4_ 10 mM, KH_2_PO_4_ 1.18 mM, glucose 11 mM and sucrose 200 mM, pH 7.4). The brains were mounted on a Microslicer (DTK-1000, Ted Pella, Redding, CA, USA) to obtain slices of 200 μm thickness. Subsequently, the hippocampi were isolated from the slices and deposited in an aCSF solution (Artificial Cerebrospinal Fluid: NaCl 120 mM, KCl 2 mM, CaCl_2_ 2 mM, NaHCO_3_ 26 mM, MgSO_4_ 1.19 mM, KH_2_PO_4_ 1.18 mM and glucose 11 mM, pH 7.4) in a water bath (34 °C, 1 h). Finally, the slices were superfused with aCSF solution and were treated with BG (20 µg/mL) and trifluoromethoxy carbonylcyanide phenylhydrazone (FCCP, 10 µM) for 3 h. In all the experiments and procedures, the solutions were constantly bubbled with a gaseous mixture of 95% O_2_/5% CO_2_. The animals used in this study were maintained according to NIH regulations and as approved by the Bioethics Committee of the University of Concepcion under approbation CEBB 667-2020.

### 2.9. Cell viability Assay

To evaluate changes in cell viability, we used 3-[4,5-dimethylthiazol-2-yl]-2,5-diphenyl tetrazolium bromide (MTT, Sigma-Aldrich, Saint Louis, MO, USA). The procedure measures the ability of mitochondria to reduce MTT salt to the insoluble compound formazan. In brief, PC-12 cells subjected to different experimental conditions were incubated for 30 min in MTT (1 mg/mL). The hippocampal slices were incubated for 1 h in MTT (0.5 mg/mL) at 37 °C. DPBS 1X (Gibco) was used to dilute MTT. Insoluble formazan formation was solubilized in 2-propanol (100 μL), and the absorbance (560 and 620 nm) was read in a 96-well plate on the NOVOstar multiplate reader (BMG Labtech, Ortenberg, Germany).

### 2.10. Spontaneous Ca^2+^ Transients

Hippocampal neurons from 10 to 11 DIV were incubated with the fluorescent probe Fluo-4 AM (5 µM) (Invitrogen) for 20 min in DPBS 1X. Then, neurons were washed with DPBS 1X and normal external solution (SEN) and observed under an inverted microscope. Fluorescence changes (ex: 480 nm; em: 520 nm) were acquired after 200 ms exposure, every 1 s, for 10 min, using an iXon + EMCCD camera (Andor, Belfast, Ireland) and Imaging Workbench 6.0 software (Indec Biosystems, Burlington, ON, Canada). Intracellular Ca^2+^ transient signals were measured after 24 h of incubation with WG or BG (10 µg/mL).

### 2.11. Inmunofluorescence

Primary cultures of hippocampal neurons were fixed with 4% paraformaldehyde for 15 min at 4 °C and then permeabilized and blocked with 10% horse serum plus 0.1% Triton X-100 for 15 min at room temperature. The samples were incubated with the following primary antibodies: synaptic vesicle-protein 2 (SV2, 1:200, mouse monoclonal, Synaptic Systems, Gottingen, Germany), brain-derived neurotrophic factor (BDNF, 1:200, rabbit, Santa Cruz Biotechnology) and microtubule-associated protein 1B (MAP1B, 1:200, goat, Santa Cruz Biotechnology, Dallas, TX, USA). The primary antibodies were incubated with the samples for 1 h at room temperature. The corresponding secondary antibodies are the following: anti-mouse (1:200, donkey, Alexa Fluor 488, Jackson Inmuno Research); anti-rabbit (1:200, Alexa Fluor 488, Jackson Inmuno Research) and anti-goat (1:200, donkey, Cy3, Jackson Inmuno Research). The secondary antibodies were incubated with the samples for 45 min at room temperature. Finally, the preparations were mounted using immunofluorescence-mounting media (Dako, Glostrup, Denmark). Images were acquired using a Nikon eclipse TS2 (China). Image processing and quantification were performed using Image J (NIH, Bethesda, MD, USA).

### 2.12. Automated Sholl Analysis

The analysis of the neuronal morphology was performed using the Bonfire program, which is a semi-automated approach to the analysis of dendrite and axon morphology that builds upon available open-source morphological analysis tools. The Bonfire program requires the use of two open-source analysis tools, the NeuronJ plugin to ImageJ and NeuronStudio 0.9.92. Neurons are traced in ImageJ 1.54f, and NeuronStudio is used to define the connectivity between neurites. Bonfire contains several custom scripts written in MATLAB 2015 (MathWorks) that are used to convert the data into the appropriate format for further analysis, check for user errors and ultimately perform Sholl analysis [26].

### 2.13. Statistical Analysis

Results are expressed as mean ± SEM. Statistical analysis was performed using one-way ANOVA and unpaired t-tests. The results * *p* < 0.05, ** *p* < 0.01 and *** *p* < 0.001 versus control, and ^+^
*p* < 0.05, ^++^
*p* < 0.01 and ^+++^
*p* < 0.001 WG versus BG were considered statistically significant. All the analyses were performed with GraphPad Prism software (GraphPad Prism, CA, USA).

## 3. Results

### 3.1. Chemical Composition of White and Black Garlic Extract from Allium ampeloprasum var. ampeloprasum

We first analyzed the GC profile of BG and compared it with WG. Table 1 presents the assigned identities of compounds obtained in a hexane fraction, an AcOEt fraction and a methanolic hydrolyzed fraction extracted with AcOEt. The data include molecular weight, molecular formula, retention time and normalized areas of compounds. First, in the n-hexane extracts, 9 alkanes were identified in WG and BG. In both, the highest normalized areas were 1-docosanol and hexadecane. An increase in all normalized areas was observed for BG, with the exception of 1-docosanol, which halved. In the hydrolyzed fraction, 9 compounds were detected: 2 carboxylic acids, 2 fatty acids, 3 aldehydes, a derivate amino acid and a phenolic compound, with several of them detected as methyl ester forms. The highest normalized areas were observed for 5-hydroxymethylfurfural (5-HMF) and 2,4-heptadienoic acid. In BG, the aldehyde increased close to 40%, and the carboxylic acid halved. Isolinoleic acid, whose normalized area was low, showed an increase close to 40% in BG, but still with a smaller signal than other compounds. In the ethyl acetate fraction, the sulphurated compounds were extracted: 21 compounds in WG and only 9 in BG. The main compound in WG was the disulfide di-2-propenyl (Di2P or DADS), while all other 20 compounds showed normalized areas smaller than 5%. DADS was reduced by 2.7 times in BG; however, well-defined increases in 2-Propenyl sulfide (DAS), 3H-1,2-Dithiole (D3T) and 4-Methyl-1,2,3-trithiolane (TTL) were observed. The other six compounds showed normalized area smaller that 5%.

### 3.2. Antioxidant Capacity of Garlic Extracts from Allium ampeloprasum

To assess the antioxidant capacity of our extracts, we employed DPPH and ABTS assays [24]. Our results showed that the BG extract had a significantly higher antioxidant capacity than WG. The inhibition of the DPPH free radical by 10 mg/mL of BG elicited 51 ± 5.7%, whereas WG showed only 12 ± 2.6% inhibition at the same concentration (Figure 1A). Additionally, the BG extract showed a significant antioxidant capacity in DPPH experiments from lower concentrations (4 mg/mL), whereas the WG extract showed a significant activity only from 10 mg/mL. The ABTS assays correlated directly with the observations made with the DPPH method. The inhibition of ABTS free radical by using 10 mg/mL of BG and WG extract reached 69.4 ± 2.3% and 21 ± 3.9%, respectively (Figure 1B). Further analysis using TEAC assays confirmed that the BG extract (10 mg/mL) had a significantly higher antioxidant capacity in comparison with the WG extract (10 mg/mL) (Figure 1C,D).

### 3.3. Effects of Garlic Extracts on Cell Viability

To evaluate the effects of the garlic extracts on cell viability, we used the PC-12 cell and MTT method. FCCP (10 µM) was used as a control of cell toxicity. Initially, we evaluated the effects of WG and BG extracts in a wide range of concentrations (0.1–1000 µg/mL). Our results showed that both WG and BG extracts did not exert any toxic actions on the cell viability (Figure 2A,B), suggesting the absence of toxicity related with the components in the extracts. To explore whether the extracts exert potential beneficial effects on the neuronal survival, the effects of the BG extract on the cell viability were tested on mouse hippocampal slice preparations. This ex vivo preparation allowed us to investigate the neural and glial viability in the context of a more physiological model of an intact brain neuronal network. Interestingly, we observed that 20 µg/mL BG extract significantly increased the cell viability (41 ± 9% over control, Figure 2C). Taken together, these results suggest that BG extract has bioactive properties in cellular models and can promote a significant stimulatory effect on the cell viability of hippocampal neurons.

### 3.4. Functional Effects of WG and BG Extracts on the Neuronal Activity of Hippocampal Neurons

To study the potential effects of WG and BG extracts on the function of the neuronal network, we analyzed spontaneous intracellular Ca^2+^ transients in cultured hippocampal neurons by using microfluorimetric techniques (Figure 3A). Spontaneous intracellular Ca^2+^ signals are commonly employed as a measurement of neuronal activity, which is related to synaptic transmission. This process involves the fusion of synaptic vesicles with the presynaptic plasma membrane and the release of neurotransmitters. Therefore, the quantification of these signals is a useful tool for the evaluation of synaptic function under our experimental conditions. Changes in the main parameters of Ca^2+^ transients (frequency and amplitude) are indicators of modifications of neuronal network connectivity or neuronal health. Our results show that the amplitude of the Ca^2+^ transients was not modified under a chronic treatment (24 h) either with WG or BG extracts (10 µg/mL) (Figure 3A,B). However, both treatments significantly increased the frequency of the spontaneous Ca^2+^ transients by nearly three-fold (control: 100 ± 7%; WG: 228 ± 6%; BG: 289 ± 4%) (Figure 3C). Interestingly, BG displayed a significantly higher increase in frequency than WG (Figure 3C). These data suggest that WG and BG extracts have neuroactive components able to strengthen the synaptic activity and the neuronal network communication.

### 3.5. The BG Extract Modifies the Synaptic Machinery and Increases the Expression of the Brain-Derived Neurotrophic Factor (BDNF)

The beneficial effects of BG and WG extracts on the cell viability and on spontaneous Ca^2+^ signals suggest the induction of complex molecular mechanisms underlying functional changes. One possibility is the generation of morphological changes in neurons that reinforce the synaptic connectivity through presynaptic and postsynaptic adjustments. We first measured the expression of the key exocytotic protein SV-2, which is a protein involved in neurotransmitter release and commonly used as a presynaptic marker. Using fluorescence microscopy (Figure 4A), we observed that the BG extract significantly enhanced the expression of the SV-2 protein (175 ± 12% over control, Figure 4B). Although the WG extract was able to increase the SV-2 expression (144 ± 14%), the change was not statistically significant over the control. We then evaluated the effects of garlic extracts on the neuronal morphology, using the cytoskeletal neuronal protein MAP1B. The images were analyzed using a semi-automated Sholl analysis, which allows the quantification of the neuronal arborization. Our results showed that neither WG nor BG extracts significantly modified the number of neuronal projections, the distance of the projections from the soma or the number of branching points (Figure 4C,D).

Our results up to this point suggested that the garlic extracts enhanced the efficacy of the synapses without detectable changes in neuronal morphology. These observations led us to think about the participation of neurotrophic factors as potential mediators of the garlic extracts’ actions in the neuronal function. A classical neurotrophin involved in neuronal survival and functional synaptic plasticity is BDNF [27]. To assess whether the garlic extracts influenced BDNF levels, we determined the expression of BDNF in hippocampal neurons using fluorescence microscopy and specific antibodies against BDNF (Figure 4E). The quantification of the BDNF immunoreactivity showed that the WG extract did not change the expression of BDNF (Figure 4F). Interestingly, the BG extract significantly enhanced the BDNF expression (BG: 132 ± 12%). Altogether, these results suggested that BG extract, but not WG, contains active molecules that promote the expression of BDNF and SV-2.

## 4. Discussion

Classically, garlic preparations have been considered beneficial foods for human health, and efforts to study and determine the beneficial effects have been a permanent focus of interest for researchers [7]. Here, we evaluated the chemical composition and initial biological evaluation of the beneficial properties of extracts of fresh and aged elephant garlic (*Allium ampeloprasum* var. *ampeloprasum*). When comparing the chemical profiles of WG and BG, we found that the aging process induced a change in the chemical composition of the elephant garlic cloves, generating a unique chemical profile, especially in terms of sulfur compounds, which probably provide a high antioxidant capacity to BG that can promote enhanced neuronal survival and neuron health. This was reinforced by the evidence that indicated an increase in the levels of the neurotrophin BDNF, a classical promotor of neuronal activity.

Regarding the chemical profile of the hydrolyzed extracts of both WG and BG, some compounds with methyl ester forms were detected, probably due to the hydrolysis conditions used (methanol and hydrochloric acid at high temperature), which allowed us to presume that the parent compounds were not methyl ester. Second, several bioactive compounds increased their levels in BG, including 5HMF (69%), DAS (17%), D3T (22%) and TTL (16%), and other compounds had areas less than 5%. These results contrast with the results obtained by Martinez-Casas and coworkers, where they observed that for the *Allium sativum* species, the aging process decreased the percentage of most sulfur compounds, with the exception of DAS and allyl methyl trisulfide, which increased at 9%, while 5HMF increased by only 4% and the compounds D3T and TTL were not present [6].

Evidence from Liu and coworkers suggested that 5-HMF may serve as a potential therapeutic agent for the treatment of Alzheimer’s disease (AD). They showed that the cognitive impairment of mice treated with intracerebroventricular injections of Aβ_1–42_ was significantly reverted by 5-HMF. These assays showed improvements in locomotor activity, Y-maze test performance and Morris water maze test performance [28]. On the other hand, sulfur compounds are recognized as important signaling molecules involved in cytoprotection against oxidative stress and inflammatory response. In the brain, hydrogen sulfide (H_2_S) and polysulfides exert multiple functions in both health and disease, including AD, Parkinson’s disease (PD), Huntington’s disease (HD), memory decline and glioma [29]. Thus, an interesting point of discussion is the increase in sulfur molecules in BG. DAS has been described as an antineoplastic and antifungal agent with additional anti-neuroinflammatory activity [30], while D3T is a potent free radical scavenger, able to activate the Nrf2 signaling pathway. In addition, D3T provided neuroprotection in an Alzheimer’s disease (AD) cellular model of N2a/APPswe cells [31]. TTL has been less studied; however, Liang and coworkers reported that it is a potent, slow-releasing H_2_S agent [32]. H_2_S acts as a neuromodulator, facilitates long-term potentiation and regulates intracellular calcium levels, which are important processes in learning and memory. Aberrant endogenous production and metabolism of H_2_S are implicated in the pathogenesis of neurodegenerative diseases. Consequently, H_2_S donors have shown beneficial therapeutic effects in neurodegenerative disease models by targeting hallmark pathological events (e.g., Aβ production in AD and neuroinflammation in PD). The results obtained from in vivo studies clearly show that H_2_S not only prevents neuronal and synaptic deterioration but also improves deficits in memory, cognition and learning. The anti-inflammatory, antioxidant and anti-apoptotic effects of H_2_S underlie its neuroprotective properties [33].

Our in vitro antioxidant studies showed that the aging process of *Allium ampeloprasum* remarkably increased its antioxidant activity (Figure 1). The antioxidant capacity of extracts of fresh and aged garlic from *Allium sativum* was determined via the elimination of 1,1-di-2-picrylhydrazyl and hydroxyl phenyl radicals, the reducing power of ferricyanide, the chelation capacity of ferrous ions, the effect of inhibition on linoleic acid peroxidation, DPPH, ABTS and FRAP [34], and it was observed that aged garlic had significantly higher antioxidant activity than fresh garlic. However, this increase in antioxidant activity has been correlated with the specific increase in total polyphenol content [6,34] and not with sulfur compounds. A study performed by Lu and coworkers compared the total polyphenol contents of *A. sativum* and *A. ampeloprasum* using the Folin–Ciocalteu procedure. It was observed that the total polyphenol content of *A. ampeloprasum* was significantly lower than that of *A. sativum* [35]. Kim and coworkers compared the antioxidant capacities of both species through an ORAC assay and observed that the antioxidant activity of *A. ampeloprasum* was significantly higher than the antioxidant activity of *A. sativum*. In addition, the increase in antioxidant activity observed via ORAC was positively correlated with the content of organosulfur compounds (R^2^ = 0.891) [22]. Considering this evidence, we could suggest that the antioxidant activity of BG is mainly due to sulfur compounds; however, future tests are needed to determine the total polyphenol content of our garlic extracts.

Subsequently, using the PC-12 cell line, we determined that BG and WG did not exert cytotoxic effects in a wide range of concentrations (Figure 2A,B). However, these experiments also showed that the garlic extracts did not promote further viability of the cells. Nevertheless, when we tested the effect of BG on mouse brain slices, a more intact neural preparation that includes neurons and glial cells organized in a tridimensional structure, we observed an increase of ≈40% in the cell viability (Figure 2C). Considering that previous reports showed that aged garlic preparations increased neuronal viability mainly due to its content of sulfur compounds [5,16], it is likely that DAS, a sulphurated compound that was increased ≈47 times in our aged *A. ampeloprasum* preparation (see Table 1), may contribute to explaining the improvement in cell viability in brain slices. Interestingly, previous reports reported that DAS exerts robust antioxidant activity, together with anti-apoptotic actions. For example, it was reported that intraperitoneal administration of DAS to Wistar rats prevented lipid peroxidation induced by gentamicin in the kidney, re-establishing the activity of antioxidant enzymes glutathione peroxidase (GPX), catalase (CAT), glutathione reductase (GR), superoxide dismutase (SOD), quinone reductase (QR) and glutathione-s-transferase (GST). Another report showed that DAS increased the expression of the erythroid 2 nuclear factor related to factor 2 (NRF2) and decreased the immunoreactivity of alpha tumor necrosis factor (TNF-alpha), suggesting a DAS-induced improvement in the cellular antioxidant activity and, at the same time, a suppression of inflammatory cytokine expression [36]. DAS also reduced caspase-3 expression and increased the expression of Bcl-2 [37]. Another study in Swiss albino mice reported that DAS oral intubation (250 µg/mouse or 500 µg/mouse) inhibited apoptosis in the liver induced by a intragastric dose of the cancerogenic compound 7,12-dimethylbenz (a) anthracene (DMBA), likely through the regulation of Bax/Bcl-2 levels in proteins and the restoration of the mitochondrial membrane potential [38]. In the same line, DAS supplementation in Swiss albino mice protected against oxidative stress produced by a intragastric dose of DMBA, restoring antioxidant enzyme levels and decreasing reactive oxygen species and lipid peroxidation [38]. Finally, DAS intraperitoneal administration in rats reduced the effects of focal cerebral ischemia induced by temporary occlusion of the middle cerebral artery. The pretreated animals with DAS showed a volume of infarct and a percentage of apoptotic cells significantly lower than the animals not treated with DAS [37]. In addition, DAS reduced caspase-3 expression and increased Bcl-2 expression [37]. Considering these reports, we suggest that the higher DAS content of our BG extracts may explain the improved antioxidant capacity and positive actions in terms of cell viability. However, a potential contribution of other sulfur compounds in our samples, such as DADS, cannot be discarded. Of note, since these beneficial actions are apparently not related to the presence of SAC or SAMC [5], our results suggest an unrecognized capability of DAS as a neuroprotective agent.

The examination of cytosolic calcium oscillations in living mouse hippocampal neurons added a major layer of complexity to the aged garlic beneficial effects. Our results indicated that BG and WG were able to significantly increase the frequency of intracellular calcium transients, but this effect was greater with BG (Figure 3A,C). Interestingly, the increase in Ca^2+^ transients correlated with a significant increase in the levels of the pre-synaptic protein SV-2 in neurons treated with BG (Figure 4A,B). Our data revealed that compounds present in garlic increased intracellular calcium transients and levels of the synaptic protein SV-2 in hippocampal neurons and reinforced the idea that the molecular composition of BG strengthens the synaptic transmission of a neuronal network. This potentiation of the calcium-mediated synaptic neurotransmission could be extrapolated and associated with more complex neural processes, such as long-term potentiation, learning and memory [39].

The mechanisms underlying the potentiation of neuronal activity by BG are so far unknown. However, a key process traditionally involved in synaptic plasticity is morphological changes at the neuronal level. An adequate neuronal morphology is closely related to correct functioning and neuronal communication [26]. It has been observed, for example, that chronic behavioral stress generates changes in the morphology of the pyramidal neurons of the CA3 region of the hippocampus and prefrontal cortex of rats by reducing the branching points and the length of the apical dendrites [40]. Our results showed that either white or black garlic preparations did not significantly alter neuronal morphology (Figure 4C). These observations contrast with a report by Moriguchi and coworkers, which showed that garlic compounds that had a thioallyl group attached to a sulfur atom had neurotrophic activity, including DAS and DADS, which were present in our preparations [5]. This discrepancy may be explained by the different incubation times used in these studies. While Moriguchi and coworkers analyzed neuronal morphology after 72 h of treatment, our experiments were performed after 24 h of incubation. In addition, Moriguchi and coworkers incubated the neurons with purified compounds [5], while we tested a garlic extract, where the synergy of different molecules can have a differential effect.

Since BG extract enhanced the synaptic efficacy without evident variations in neuronal morphology, we explored a potential participation of neurotrophic factors as potential chemical mediators. Within the neurotrophins, BDNF has been characterized as a key factor involved in synaptic plasticity processes. For example, BDNF has been reported to enhance excitatory postsynaptic currents and increase the intracellular calcium concentration [27]. Of note, our results showed that only BG was able to increase the immunoreactivity of BDNF in hippocampal neurons (Figure 4E,F), suggesting an upregulation of BDNF expression triggered by BG chemical compounds. This finding supports the idea that BG improves neuronal survival and synaptic connectivity through increased BDNF signaling. Since the WG preparation was unable to modify the BDNF immunoreactivity in our experiments, it is possible to suggest that the chemical composition of BG extracts possesses key molecules that enhance BDNF expression. The higher content of DAS in BG suggests a pivotal role of this specific compound in the processes of synaptic plasticity. The potential beneficial role of DAS contrasts with observations from other research groups, which found that DADS, another garlic sulfur compound, decreased the BDNF levels in the hippocampus, promoting defects in memory and neurocognitive functions [41]. From these results, it is possible to suggest that the strong decrease in the DADS content triggered by the aging process of *A. ampeloprasum* (Table 1) allowed the increase in BDNF in hippocampal neurons, mediated by other sulfur components. In this regard, and considering DAS and DADS as exceptions, how the biological activity of other sulfur molecules increases in the aging process of *A. ampeloprasum*, such as TTL, 5-methyl-1,2,3,4-tetrathiane, 4H-1,2,3-trihiine, D3T and di-2-propenyl trisulfide, has not yet been explored. Further experiments are needed to dissect the individual biological properties of this set of compounds.

In conclusion, our findings provide novel information about the chemical composition and functional properties of aged elephant garlic, a food preparation obtained from *A. ampeloprasum,* an endemic variety of garlic found in Chiloe Island in Chilean Patagonia. The unique chemical profile of aged elephant garlic and its beneficial neuroactive properties provide the initial scientific bases to foster its use as a functional food containing a mixture of bioactive compounds that may contribute to brain health and well-being.

## Figures and Tables

**Figure 1 foods-12-03968-f001:**
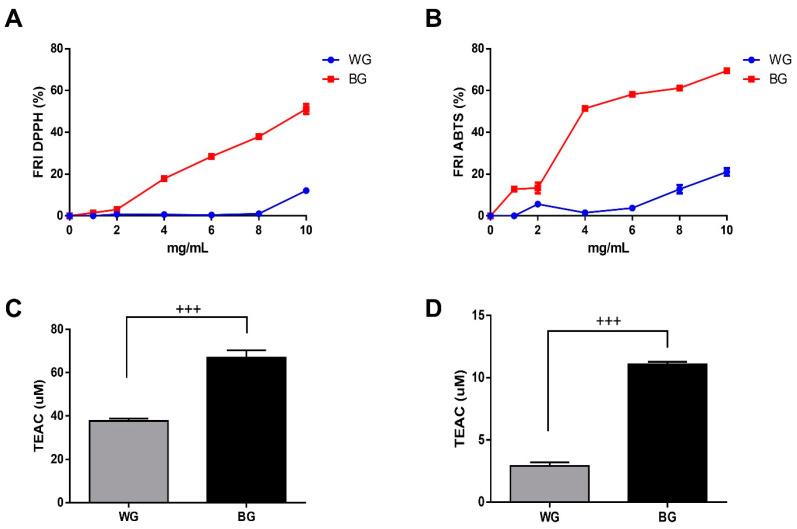
Comparative antioxidant activity of WG and BG extracts. (**A**) Inhibition of the free radical DPPH elicited by a range of BG and WG extract concentrations (0–10 mg/mL). (**B**) Inhibition of the free radical ABTS elicited by the same experimental conditions used in (**A**). (**C**,**D**) Comparison of the antioxidant capacity of WG (10 mg/mL) and BG (10 mg/mL) taken from panels (**A**) and (**B**), respectively, expressed as Trolox equivalent antioxidant capacity (TEAC). FRI: Free radical inhibition. (*n* = 3; *N* = 9). ^+++^
*p* < 0.001 WG versus BG.

**Figure 2 foods-12-03968-f002:**
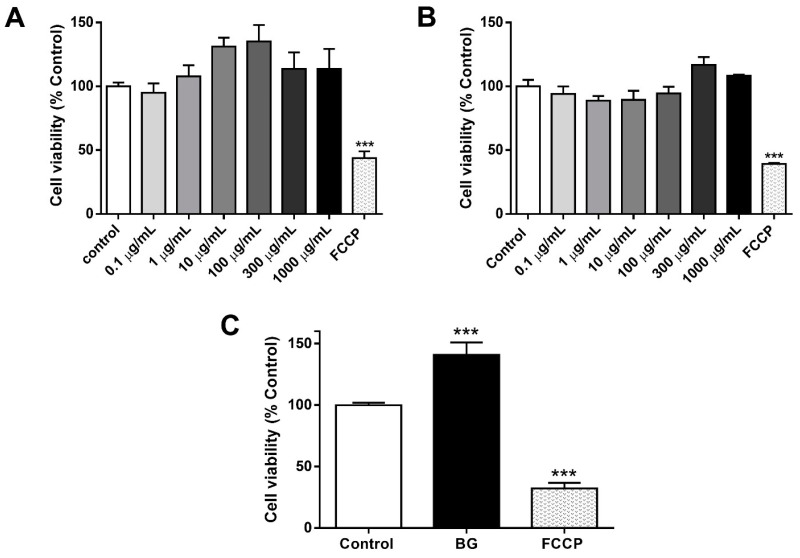
Effects of WG and BG extracts on cell viability. (**A**) Cell viability of PC-12 cells treated (24 h) with different concentrations of WG extract (0.1–1000 μg/mL). (**B**) Cell viability of PC-12 cells treated (24 h) with different concentrations of BG extract (0.1–1000 μg/mL) using MTT method. (**C**) Viability evaluation of mouse hippocampal slices under control conditions and after 3 h of incubation with BG extract (20 μg/mL). FCCP 10 µM was used as a positive control cell toxicity (*n* = 3, *N* = 9) *** *p* < 0.001 versus control.

**Figure 3 foods-12-03968-f003:**
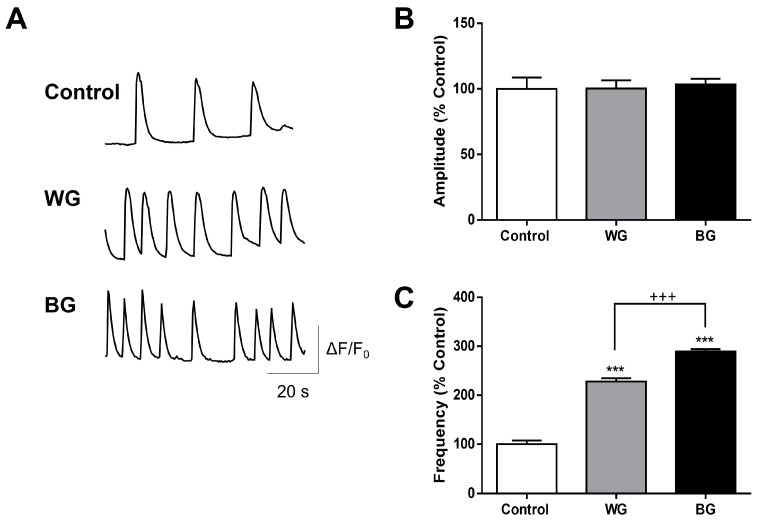
The frequency of spontaneous Ca^2+^ transients of living hippocampal neurons are enhanced by WG and BG extracts. (**A**) Representative recording traces of spontaneous Ca^2+^ signals on hippocampal neurons in control conditions (vehicle-treated) or treated with WG and BG extracts (10 mg/mL). (**B**) Evaluation of effect of extracts on the Ca^2+^ transients’ amplitude (F/Fo represented as % of control). (**C**) Evaluation of the frequency (events/min; shown as percentage over control) under the experimental conditions described above. (*n* = 3; *N* = 40); *** *p* < 0.001 *versus* control; +++ *p* < 0.001 WG versus BG.

**Figure 4 foods-12-03968-f004:**
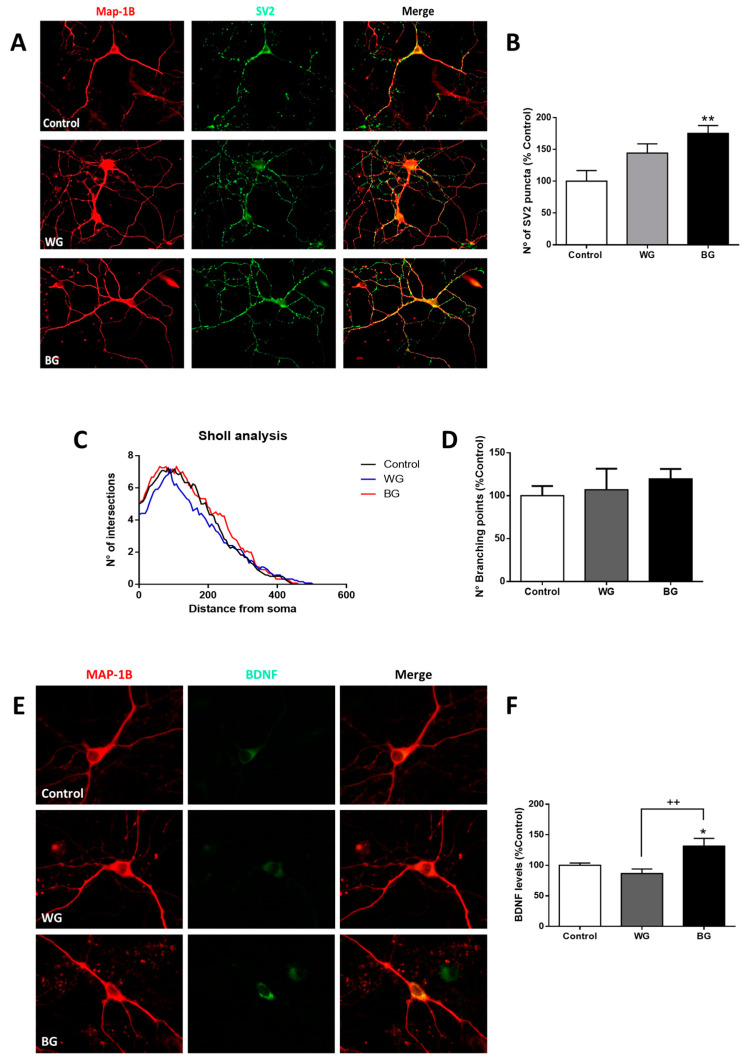
BG extract increased the expression of SV-2 and BDNF. (**A**) Representative fluorescence microscopy images of hippocampal neurons (10 DIV) treated for 24 h with BG or WG (10 mg/mL) extracts. The image shows a neuronal marker MAP1B (red, left panel), synaptic protein SV-2 (green, middle panel) and the merge of both channels (right panel). (**B**) Quantification of the number of SV-2 protein puncta (*n* = 3; *N* = 40). (**C**) Number of intersections and neuronal projections (distance) obtained with Sholl analysis. (**D**) Analysis of the number of neuronal branching points shown in (**C**). Differences were not significant. (**E**) Representative fluorescence microscopy images show hippocampal neurons with MAP1B neuronal marker (red, left panel), BDNF immunoreactivity (green, middle panel) and merge of both channels (right panel) under the same treatments described in (**A**). (**F**) Graphs show quantification of BDNF immunoreactivity. Values represent mean ± SEM as percentage of control (*n* = 3, *N* = 9). * *p* < 0.05, ** *p* < 0.01 *versus* control; ++ *p* < 0.01 WT versus BG.

**Table 1 foods-12-03968-t001:** Chemical profile of fatty acid, aromatic and sulphurated compounds of white and black garlic extract from *Allium ampeloprasum* var. *ampeloprasum*.

		White Garlic	BlackGarlic	
Retention Time(Tr, min)	Compound	Area %	Area %	Structure
	Liquid–liquid extractionn-hexane			
12.019	Heptadecane	3.09	6.44	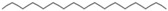
13.051	Hexadecane	13.33	28.6	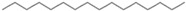
13.113	Octadecane	2.42	4.83	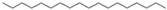
14.414	1-iodooctadecane	3.99	4.91	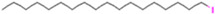
15.071	1-tetracosene	2.68	7.85	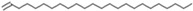
16.066	Heneicosane	3.3	4.56	
16.435	1-iodohexadecane	1.68	4.58	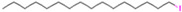
16.935	1-docosanol	68.24	35.04	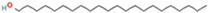
18.624	Hentriacontane	1.28	3.2	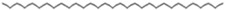
	Acid hydrolysis—ethyl acetate			
4.796	Hepta-2,4-dienoic acid, methyl ester	43.11	27.41	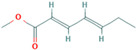
6.233	5-Hydroxymethyl-2-furaldehyde	50.23	69.1	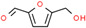
7.141	5-Acetoxymethyl-2-furaldehyde	0.24	0.05	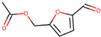
8.11	DL-Proline, 5-oxo-, methyl ester	0.13	0.08	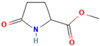
9.297	Citric acid, trimethyl ester	1.53	0.33	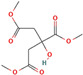
14.422	Hexadecanoic acid, methyl ester	0.61	0.37	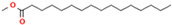
15.171	5,5′-Oxybis(5-methylene-2-furaldehyde)	1.55	0.24	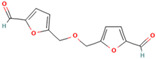
16.055	9,11-Octadecadienoic acid, methyl ester, (E, E)-	1.6	2.17	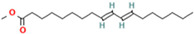
18.913	p-Cresol, 2,2′-methylenebis [6-tert-butyl-	1	0.25	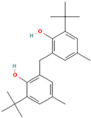
	Sulphurated compounds (HS-SPME)			
5.297	2-Propenyl sulfide	0.37	17.46	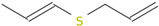
6.581	Disulfide, methyl 2-propenyl	0.52	-	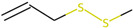
7.538	3H-1,2-Dithiole-3-thione	4.19	22.86	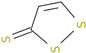
10.526	Disulfide, di-2-propenyl	83.42	29.74	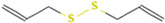
10.758	Disulfide, 1-methylethyl 2-propenyl	1.74	-	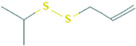
10.905	Disulfide, 1-propenyl 2-propenyl	3.51	-	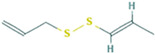
11.694	Trisulfide, methyl 2-propenyl	0.36	0.43	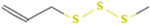
12.041	4-Methyl-1,2,3-trithiolane	0.58	16.1	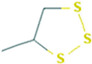
12.788	3-Vinyl-3,6-dihydro-1,2-dithiine	2.13	3.19	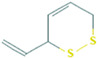
13.009	4H-1,2,3-Trithiine	0.52	3.03	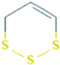
13.272	2-Vinyl-4H-1,3-dithiine	0.18	-	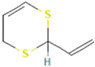
15.082	Trisulfide, di-2-propenyl	0.8	2.79	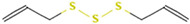
15.314	Trisulfide, 2-propenyl propyl	0.14	-	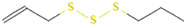
15.577	Trisulfide, 1-propenyl, 2-propenyl	0.23	-	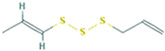
16.376	5-Methyl-1,2,3,4-tetrathiane	0.6	4.4	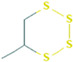
16.713	1,3-Dithiole-2-thione	0.24	-	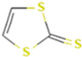
17.218	Disulfide, methyl 1-(methylthio) propyl	0.05	-	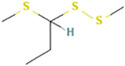
19.575	Trisulfide, 2-propenyl propyl	0.2	-	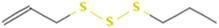
21.29	4-Methyl-1,2,3-trithiolane	0.17	-	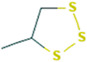
23.353	1,3-Butadiene, 3-methyl-1,1-bis (methylthio)-	0.03	-	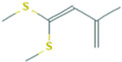
23.889	Trisulfide, 2-propenyl 2-(2-propenylthio) propyl	0.01	-	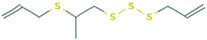
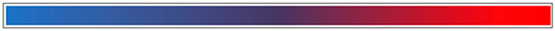 Scale Percentual (%) 1–100

## Data Availability

Data is unavailable due to privacy.

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
