# Peer review of "Elephant Black Garlic’s Beneficial Properties for Hippocampal Neuronal Network, Chemical Characterization and Biological Evaluation"

_foods, 2023, doi:10.3390/foods12213968_

Round 1

Reviewer 1 Report

The study evaluated the in vitro antioxidant and neuroprotective activities of aged extracts from garlic. The experiments were well-performed and the manuscript was well-written. It needs some minor revisions.

1) All abbreviations should be introduced at the first time of appearance, such as "DPPH" and "ABTS" in Abstract.

2) Line 115: For an aging process, a temperature of 60 °C, humidity of 95% and a period of 20 days were applied. Any reference?

3) For inmunofluorescence assay, why were only SV2, BDNF and MAP1B tested?

4) The most limitation of this work is that we don't know whether these antioxidant compounds could pass the blood brain barrier or may be degraded at the gastrointestinal tract. A feeding experiment may resolve this issue. Or at least, these problems should be discussed.

Pay attention to the subscript, such as Na2CO3 in Line 136, Na2SO4 in Line 138, and CO2 in Line 216. "ºC" should be "°C", which should be corrected throughout the text.

Author Response

REVIEWER 1

The study evaluated the in vitro antioxidant and neuroprotective activities of aged extracts from garlic. The experiments were well-performed and the manuscript was well-written. It needs some minor revisions.

1) All abbreviations should be introduced at the first time of appearance, such as "DPPH" and "ABTS" in Abstract.

- Corrected

2) Line 115: For an aging process, a temperature of 70°C, humidity of 90% and a period of 20 days were applied. Any reference?

-Kimura, S.; Tung, Y.-C.; Pan, M.-H.; Su, N.-W.; Lai, Y.-J.; Cheng, K.-C. Black garlic: A critical review of its production, bioactivity, and application. Journal of food and drug analysis 2017, 25, 62-70.

-Jing, H. Black garlic: Processing, composition change, and bioactivity. EFood 2020, 1, 242-246.

Booth Has been incorporated.

3) For inmunofluorescence assay, why were only SV2, BDNF and MAP1B tested?

We used the synaptic markers that we used to use to evaluate the changes in the synaptic organization on neural network under the experimental conditions, we have wide experience evaluating changes in these markers in our previous works.

4) The most limitation of this work is that we don't know whether these antioxidant compounds could pass the blood brain barrier or may be degraded at the gastrointestinal tract. A feeding experiment may resolve this issue. Or at least, these problems should be discussed.

Currently, there are no specific studies on the pharmacokinetics of the compounds present in our garlic extract. However, DADS and DAS are compounds derived from SAC, one of the main components present in garlic described in the literature (Martínez-casas et al. 2017).

In rats, oral administration of SAC shows a three-phase concentration profile: two very rapid phases (absorption and distribution) followed by a slow elimination phase, like the pharmacokinetics of SAC by intravenous administration (Nagae et al. 1994; Yan and Zeng, 2005). The oral bioavailability of SAC at a dose of 100 mg/kg is 91% (Yan and Zeng, 2005). In addition, a pharmacokinetic study of SAC in humans has been performed by oral administration of a garlic preparation. The half-life of SAC in humans after oral administration was greater than 10 h, and the elimination time was greater than 30 h (Kodera et al 2002). The total content of SAC in the blood of volunteers at Tmax is about 450 μg (Tmax content, 23 ng/mL plasma; body weight: 65 kg; whole blood volume: 1/3 of body weight), suggesting high bioavailability in humans (Kodera et al 2002).

After oral intake, SAC is absorbed in the gastrointestinal tract and can be detected in various tissues up to 8 h after dosing (Nagae et al. 1994; Yan and Zeng, 2005). In rats, after a single oral dose of SAC (50 mg/kg), the highest peak concentration is observed in the kidney (Cmax= 65.7 mg/kg at 10 min), while in brain the Cmax is 26.7 mg/kg at 10 min and a half-life (T1/2) of 1.2 h, indicating that SAC can cross the blood-brain barrier. N-acetyl-SAC has been identified as a metabolite of SAC in the urine of rats, dogs, and humans (Jandke and Spiteller ,1987; Nagae et al. 1994). These data suggest that SAC is taken up and metabolized to N-acetyl-SAC by N-acetyltransferase, which is found primarily in the liver and kidney. However, it has been shown that SAC is almost eliminated from the liver but is in high concentration in the kidney. Therefore, it can be speculated that SAC may be transformed into N-acetyl-SAC by N-acetyltransferase in the liver, and then a portion of N-acetyl-SAC is transformed into SAC by acylase in the kidney, followed by its reabsorption (Nagae et al. 1994).

Additionally, a recent study developed the use of nanostructured lipid carriers (NLC) to enhance the permeation of compounds present in a garlic oil across the blood-brain barrier (BBB). The size of garlic NLC was 136.8±0.56 nm and was able to cross a BBB in vitro mimicked with mouse microvascular endothelial cells (bend.3) co-cultured with astrocytes. Thus, at present, there are possibilities for bioactive molecules present in garlic to be considered in brain health (Dana et al. 2021).

Martínez-Casas, L., Lage-Yusty, M., & López-Hernández, J. (2017). Changes in the aromatic profile, sugars, and bioactive compounds when purple garlic is transformed into black garlic. Journal of agricultural and food chemistry, 65(49), 10804-10811.

  1. Kodera, A. Suzuki, O. Imada et al. (2002). “Physical, chemical, and biological properties of S-allylcysteine, an amino acid derived from garlic,” Journal of Agricultural and Food Chemistry, vol. 50, no. 3, pp. 622–632.
  2. K. Yan and F. D. Zeng (2005). “Pharmacokinetics and tissue distribution of S-allylcysteine,” Asian Journal of Drug Metabolism and Pharmacokinetics, vol. 5, no. 1, pp. 61–69.
  3. Nagae, M. Ushijima, S. Hatono et al. (1994) “Pharmacokinetics of the garlic compound S-allylcysteine,” Planta Medica, vol. 60, no. 3, pp. 214–217.
  4. Jandke and G. Spiteller (1987) “Unusual conjugates in biological profiles originating from consumption of onions and garlic,” Journal of Chromatography, vol. 421, no. 1, pp. 1–8.

-Dana, P., Yostawonkul, J., Chonniyom, W., Unger, O., Sakulwech, S., Sathornsumetee, S., & Saengkrit, N. (2021). Nanostructured lipid base carrier for specific delivery of garlic oil through blood brain barrier against aggressiveness of glioma. Journal of Drug Delivery Science and Technology64, 102651.

Comments on the Quality of English Language

Pay attention to the subscript, such as Na2CO3 in Line 136, Na2SO4 in Line 138, and CO2 in Line 216. "ºC" should be "°C", which should be corrected throughout the text.

- Corrected

Reviewer 2 Report

Review report:

The article Titled “Elephant Black Garlic Beneficial Properties on Hippocampal 2 Neuronal Network, Chemical Characterization and Biological 3 Evaluation” is scientifically sound with interesting research work, but there are few concerns which should must be addressed.

1. In abstract please mention the timeline of aged extract.

2. There are numerous grammatical errors and sentences with lack of sense, therefore the article must be revised for English language with the help of expert of the language.

3. Line 118: Please mention the strength of methanol (% purity).

4. Line 140: Please write the full form in the headings

5. The authors did not differentiate between the fresh and aged extracts.

6. Lines 415-419: must be rephrased.

7. Line 427-428: please provide proper reference for your statement.

8. Line 445: Please remove the word “other” or specify the other diseases.

9. All the abbreviations must be defined at their first use.

10. The first three paragraphs of discussion section are seeming like a literature review, although the information is interesting and meaningful but need a vigorous correlation with the results of the current study.

11. Line 461: Please rephrase “in our hands”.

12. Line 469-470: improper justification, because the garlic collected from different regions and passed through various kind of processing might present different scenario of polyphenolic composition, so it would be better to evaluate the polyphenols in this study too.

13. Line 480-481: must be rephrased.

14. Overall, there should be proper correlation of the current results with literature.

15. Why the authors did not carry out the antioxidant evaluation through power reducing assays?

16. why the authors used stirring method for preparation of the extract, as this may cause damage of certain secondary metabolites.

Moderate English editing required.

Author Response

REVIEWER 2

The article Titled “Elephant Black Garlic Beneficial Properties on Hippocampal 2 Neuronal Network, Chemical Characterization and Biological 3 Evaluation” is scientifically sound with interesting research work, but there are few concerns which should must be addressed.

  1. In abstract please mention the timeline of aged extract.

-Done

  1. Line 118: Please mention the strength of methanol (% purity).

-Methanol HPLC grade, ≤99.9 % purity, corrected

  1. Line 140: Please write the full form in the headings

- Corrected

  1. Lines 415-419: must be rephrased.

-Done

  1. Line 427-428: please provide proper reference for your statement.

-Done

  1. Line 445: Please remove the word “other” or specify the other diseases.

-Done

  1. All the abbreviations must be defined at their first use.

- Corrected

  1. Line 461: Please rephrase “in our hands”.

-Done

  1. Line 469-470: improper justification, because the garlic collected from different regions and passed through various kind of processing might present different scenario of polyphenolic composition, so it would be better to evaluate the polyphenols in this study too.

-Corrected

  1. Line 480-481: must be rephrased.

-Done

  1. Why the authors did not carry out the antioxidant evaluation through power reducing assays?

The DPPH and ABTS assays were used for the analysis of antioxidant activity because of their low cost and availability, and because they are the assays we have standardized in our laboratory. An advantage of the ABTS assay is that it allows us to directly contact the antioxidant agent with a radical and allows measurement at different wavelengths. Furthermore, ABTS•+ is soluble in both water and organic solvents, which enables the antioxidant capacity of both hydrophilic and lipophilic compounds to be determined with the same basic methodology (Optiz et al 2014). A disadvantage of using the Ferric reducing antioxidant power (FRAP) technique is that the reaction must be carried out at acidic pH (3.6) (Zhong and Shahidi, 2015) and the composition of the extract may be affected, whereas DPPH and ABTS can be used in a wider pH range. We believe that the information provided by the analyses performed meets the objectives proposed in this research and allowed us to observe differences between fresh and aged garlic extract of Allium ampeloprasum species. In addition, recent studies of plant extracts have recurrently used the DPPH and ABTS methods for the analysis of antioxidant activity (Vinci et al. 2022). However, for future research we will consider the evaluation of antioxidant capacity through FRAP, Oxygen Radical Absorbance Capacity (ORAC) and/or Reducing power (RP) techniques because they are widely used to evaluate the antioxidant activity of foods.

Opitz, S. E., Smrke, S., Goodman, B. A., & Yeretzian, C. (2014). Methodology for the measurement of antioxidant capacity of coffee: A validated platform composed of three complementary antioxidant assays. In Processing and Impact on Antioxidants in Beverages (pp. 253-264). Academic Press.

Zhong, Y., & Shahidi, F. (2015). Methods for the assessment of antioxidant activity in foods. In Handbook of antioxidants for food preservation (pp. 287-333). Woodhead Publishing.

Vinci, G., D’Ascenzo, F., Maddaloni, L., Prencipe, S. A., & Tiradritti, M. (2022). The influence of green and black tea infusion parameters on total polyphenol content and antioxidant activity by ABTS and DPPH assays. Beverages8(2), 18.

  1. why the authors used stirring method for preparation of the extract, as this may cause damage of certain secondary metabolites.

-Magnetic stirred extraction is suitable for the recovery of a wide range of compounds and is recognized mo0k as a versatile and efficient extraction technique for secondary metabolites from plants. Frequent stirring during maceration facilitates extraction by two processes: (1) promoting diffusion, (2) separating concentrated solution from the sample surface by adding new solvent to the menstruum to increase extraction yield. (Jackson, 2008; Srivastava et al., 2021).

Jackson RS. Wine Science: Principles and Applications. Academic Press; 2014.

Srivastava N, Singh A, Kumari P, Nishad JH, Gautam VS, Yadav M, et al. Advances in extraction technologies: isolation and purification of bioactive compounds from biological materials. En: Natural Bioactive Compounds. Elsevier; 2021. p. 409–33.

Reviewer 3 Report

The manuscript looks interesting from the point of view of practical significance.

However, I have a few comments.

White garlic is aged at 60 degrees and 95% humidity - is there no growth of any thermotolerant bacteria or microfilamentous fungi? Perhaps the changes in the composition are related to any contamination of raw materials?

The data on the normalized amount of identified substances in the table looks very cumbersome and difficult to perceive. I can recommend the authors to visualize them in the form of a diagram or heat map. Moreover, the structures of compounds can be added to Table 1 for a better understanding.

The data in Figure 2 raises questions. At first glance, it seems to me that WG extract in concentrations of 10 and 100 mkg/ml causes an increase in cell viability. Does the statistics say that these differences are insignificant? The same question applies to the extract of BG at a concentration of 300 mkg/ml. Accordingly, I do not understand why only one extract was used for the study using hippocampal slices and at only one concentration.

I also have questions about the interpretation of data on the entry of calcium into cells. I think the authors need to analyze this data more carefully. Under what other conditions can an increased calcium intake be observed, and is it really good? The MTT test has its limitations and is not always relevant, so it is impossible to unambiguously judge the well-being of cells only using the mtt test. Perhaps some pores are formed that contribute to the entry of calcium? Then you need to evaluate the viability of cells by LDH output, for example.

The name of the section "3.4. Functional effects of WG and BG extracts on the neuronal activity of hippocampal neurons" seems to me incorrect. The study of calcium intake can hardly be considered a test for functional activity. It is obvious that the assessment of functional activity is more correctly performed by methods of electrophysiology.

Author Response

REVIEWER 3

The manuscript looks interesting from the point of view of practical significance.

However, I have a few comments.

White garlic is aged at 60 degrees and 95% humidity - is there no growth of any thermotolerant bacteria or microfilamentous fungi? Perhaps the changes in the composition are related to any contamination of raw materials?

The products produced by the agricultural company Melimei have undergone sanitary registration, are of certified edible quality and there is no history of bacterial and/or fungal contamination.

It has been widely described in the literature that the organosulfur compounds present in garlic have high antibacterial activity, even against multidrug-resistant strains (Bhatwalkar et al. 2021) and antifungal activity (El-Saber Batiha et al. 2020), which would prevent the growth of microorganisms.

On the other hand, the compounds we report in Table 1 are characteristic of the Maillard reaction for Allium sp (Kimura et al. 2017, Martinez-Casas et al. 2017).

Bhatwalkar, S. B., Mondal, R., Krishna, S. B. N., Adam, J. K., Govender, P., & Anupam, R. (2021). Antibacterial properties of organosulfur compounds of garlic (Allium sativum). Frontiers in Microbiology, 12, 1869.

El-Saber Batiha, G., Magdy Beshbishy, A., G. Wasef, L., Elewa, Y. H., A. Al-Sagan, A., Abd El-Hack, M. E., ... & Prasad Devkota, H. (2020). Chemical constituents and pharmacological activities of garlic (Allium sativum L.): A review. Nutrients12(3), 872.

Kimura S, Tung YC, Pan MH, Su NW, Lai YJ, Cheng KC. Black garlic: A critical review of its production, bioactivity and application. Yao Wu Shi Pin Fen Xi. 2017;25:62–70.

Martínez-Casas L, Lage-Yusty M, López-Hernández J. Changes in aromatic profile, sugars and bioactive compounds when purple garlic is transformed into black garlic. J Agric Food Chem. 2017;65:10804–10811. doi: 10.1021/acs.jafc.7b04423.

The data on the normalized amount of identified substances in the table looks very cumbersome and difficult to perceive. I can recommend the authors to visualize them in the form of a diagram or heat map. Moreover, the structures of compounds can be added to Table 1 for a better understanding.

-Table 1 edited

The data in Figure 2 raises questions. At first glance, it seems to me that WG extract in concentrations of 10 and 100 mkg/ml causes an increase in cell viability. Does the statistics say that these differences are insignificant? The same question applies to the extract of BG at a concentration of 300 mkg/ml. Accordingly, I do not understand why only one extract was used for the study using hippocampal slices and at only one concentration.

According with the statistical tests, non-differences between groups respect to the control were observed. We tested only BG on hippocampal slices due that this extract is our focus of interest and additionally  under 3R criteria, the experiment with WG appears to be  non necessary to the final interpretation of these results.

I also have questions about the interpretation of data on the entry of calcium into cells. I think the authors need to analyze this data more carefully. Under what other conditions can an increased calcium intake be observed, and is it really good?

Spontaneous Calcium transients as well synaptic currents  (measured by patch clamp tehcniques), represent an strong evidences about the neural network activity, and have  a tight correlation, when the neural network increase their activity, the calcium spikes increase showing to be a reflex of the synaptic healty or alterations, we have demonstrated these correlations in our previous works.

The MTT test has its limitations and is not always relevant, so it is impossible to unambiguously judge the well-being of cells only using the mtt test. Perhaps some pores are formed that contribute to the entry of calcium? Then you need to evaluate the viability of cells by LDH output, for example.

The conductance of calcium mediated by pore formation or  through a dysregulated mechanism shown a completely different kinetics, with a sustained increase to elicit a irreversible plateau. The spikes of calcium as a transient event shown that the postsynaptic neurons have been stimulated by a quantum of neurotransmitter, and the cell response activating the control mechanisms to store the calcium quickly. A  non healty cell is unable to manage the cytosolic calcium increase correctly. Mtt did not induce any alteration on  synaptic calcium signalling in our experiments.

The name of the section "3.4. Functional effects of WG and BG extracts on the neuronal activity of hippocampal neurons" seems to me incorrect. The study of calcium intake can hardly be considered a test for functional activity. It is obvious that the assessment of functional activity is more correctly performed by methods of electrophysiology.

As we mentioned upper, we have wide evidence in our lab, that the electrophysiology activity have a strong correlate with calcium transient activity.

Sáez-Orellana F, Godoy PA, Bastidas CY, Silva-Grecchi T, Guzmán L, Aguayo LG, Fuentealba J. ATP leakage induces P2XR activation and contributes to acute synaptic excitotoxicity induced by soluble oligomers of β-amyloid peptide in hippocampal neurons. Neuropharmacology. 2016 Jan;100:116-23. doi: 10.1016/j.neuropharm.2015.04.005. Epub 2015 Apr 17. PubMed PMID: 25896766.

Fuentealba J, Dibarrart A, Saez-Orellana F, Fuentes-Fuentes MC, Oyanedel CN, Guzmán J, Perez C, Becerra J, Aguayo LG. Synaptic silencing and plasma membrane dyshomeostasis induced by amyloid-β peptide are prevented by Aristotelia chilensis enriched extract. J Alzheimers Dis. 2012;31(4):879-89. doi: 10.3233/JAD-2012-120229. PubMed PMID: 22728896.

Fuentealba J, Dibarrart AJ, Fuentes-Fuentes MC, Saez-Orellana F, Quiñones K, Guzmán L, Perez C, Becerra J, Aguayo LG. Synaptic failure and adenosine triphosphate imbalance induced by amyloid-β aggregates are prevented by blueberry-enriched polyphenols extract. J Neurosci Res. 2011 Sep;89(9):1499-508. doi: 10.1002/jnr.22679. Epub 2011 Jun 6. PubMed PMID: 21647937.

Round 2

Reviewer 2 Report

The manuscript can be accepted for possible publication now.

Author Response

thank you very much

Reviewer 3 Report

The authors slightly revised the manuscriptÑŽ

However, I cannot consider the authors' response satisfactory.

"The conductance of calcium mediated by pore formation or  through a dysregulated mechanism shown a completely different kinetics, with a sustained increase to elicit a irreversible plateau. The spikes of calcium as a transient event shown that the postsynaptic neurons have been stimulated by a quantum of neurotransmitter, and the cell response activating the control mechanisms to store the calcium quickly. A  non healty cell is unable to manage the cytosolic calcium increase correctly. Mtt did not induce any alteration on  synaptic calcium signalling in our experiments."

This maxim is not supported by the drawings given in the manuscript. Figure 3 does not contain a long enough graph to illustrate the effects of the extracts. Moreover, the authors did not use any positive and negative controls to confirm the controlled calcium dynamics that ensure healthy cellular functional activity. MTT data are not sufficient to confirm cell integrity.

Author Response

So sorry for our previous imprecise answer. We want to make reference to our previous experience where we have demonstrated the correlation between Calcium transientes and synaptic electrohpysiological activity, and additionally how amyloid beta peptide aggregated induce aberrant Ca increases and its impact on synaptic activity. we would like to mention the figs1-4 of follow paper doi: 10.1016/j.neuropharm.2015.04.005. that shown our previous and actual comments.
